## Modelling extreme discharge response to several geostatistically interpolated rainfall using very sparse raingage data

Sarann Ly<sup>1</sup>, Catherine Sohier<sup>2</sup>, Catherine Charles<sup>3</sup>, Aurore Degré<sup>2</sup>

<sup>1</sup> Institute of Technology of Cambodia, Department of Rural Engineering, Russian Federation Boulevards, PO Box 86, Phnom 5 Penh, Cambodia

<sup>2</sup> Univ. Liège, Gembloux Agro-Bio Tech, Soil Water Plant Exchange, Passages des Déportés, 2, 5030 Gembloux, Belgium <sup>3</sup> Univ. Liège, Gembloux Agro-Bio Tech, Ecosystems-Atmosphere Exchanges, avenue de la Faculté d'Agronomie 8, 5030 Gembloux, Belgium

Correspondence to: Sarann Ly (ly.sarann@itc.edu.kh)

- Abstract. This study presents modelling work of extreme discharge response to rainfall inputs interpolated by geostatistical 10 approaches. Multivariate geostatistics are used by incorporating elevation as external data to improve the rainfall prediction. Thirty year daily rainfall in the Ourthe and Ambleve nested catchments, located in the Ardennes hilly landscape in the Walloon region, Belgium are interpolated and then used as inputs for a distributed physically-based hydrological model (EPIC-GRID). The effect of different raingage densities and particularly the effect of the raingage positions for very sparse raingage data used
- for rainfall interpolation, on extreme flow is analysed. We propose an index that can illustrate the quality of the raingage 15 distribution with respect to the calculation of extreme discharge. In high elevation sub-catchment, we found that the multivariate geostatistics can significantly improve the rainfall prediction to produce very good simulated peak discharge. In the low elevation sub-catchment and the low raingage density, our results indicated that the Universal Kriging (UNK) is not appropriate. The IDW, Ordinary Kriging (ORK) and Ordinary Cokriging (OCK) methods provide generally good performance.
- The Thiessen polygon (THI) and Kriging with External Drift (KED) provide good performance for the whole catchment but 20 less good for sub-catchments. The position of the raingages is the key factor for rainfall interpolation, particularly in the datascarce region. UNK and KED methods are the most sensitive.

## **1** Introduction

25

Peak discharge at a particular return period is usually required for design of water-related structures and water resource management. This is usually called "design discharge" which is commonly computed using long historical measured data at the outlet of a catchment (Rao and Hamed, 2000). However, most catchments of the world, especially in developping countries are un-gauged or poorly gauged. This can be difficult to get the correct estimate of design discharge. As a result, transportation systems and other essential elements of our hydrologic infrastructure may continue to be unreasonably vulnerable to flooding, loss of life and damage of property.

One way to determine the design discharge is to use a hydrological model to simulate discharge using rainfall which is a basic input among other inputs like Digital Elevation Model (DEM), land use, crop growth, agricultural practices, soil, and other climate parameters. Due to the great variability of rainfall in space and time, it is best to record it continuously by raingages or weather stations. Moreover, the rainfall input as other climatic parameters should be prepared by a preliminary spatial

- interpolation to get the spatial rainfall, prior to modeling. Different interpolation methods of rainfall can potentially lead to differences in the simulated peak discharge. The number and position of raingages used for rainfall interpolation can also affect the simulation results. Especially for very sparse raingage data, the discharge modelled using these raingages can lead to great risk of water-related projects. However, there is no definite rule as to how many rain gauges are needed for a fully ungauged basin (Chen et al., 2008).
- A number of investigations use a range of techniques for spatial interpolation of rainfall. The spatial interpolation methods vary in their assumptions, deterministic or geostatistical characteristics, and local or global perspective (Moral, 2010). Deterministic methods such as Inverse Distance Weighting (IDW) have been practised in many studies (Dirks et al., 1998; Lloyd, 2005). Although IDW is an objectively simple technique which offers adaptable weights for rational local interpolations, the selection of the weighting function is subjective and no measure of error is provided (Webster and Oliver,
- 2007). In principle, IDW can not clearly give a description of climatic condition while elevation extrapolation is considered necessary (Tobin et al., 2011). Consequently, the advanced application and development of multivariate geostatistics like Kriging with External Drift (KED) or Ordinary Cokriging (OCK) using several co-variables were recommended in the literature (Goovaerts, 2000; Lloyd, 2005). The geostatistical approaches remain frequently recognized to have certain advantage above the deterministic procedures (Goovaerts, 2000). An important gain of kriging is unbiased predictions with
- minimum variance and the spatial correlation between the data observed at diverse weather stations or raingages (Moral, 2010). As well as providing the prediction error, another advantage of geostatistics over deterministic approaches is the possibility of adding secondary or auxiliary data to the main data. As existing devices such satellites, radar, microwave links, etc., are available for certain industrialized countries, the data from those devices can be used to recover rainfall prediction (Haberlandt, 2007; Schuurmans et al., 2007; Velasco-Forero et al., 2009; Verworn and Haberlandt, 2011; Schiemann et al., 2011). Satellite
- data (TRMM) was used as co-variable for rainfall interpolation in India (Wagner et al., 2012). In the countries where the modern instruments are not accessible, altitude, particularly extracted from a digital elevation model (DEM), is an widely available data which can be used to integrate into multivariate geostatistics of precipitation (Goovaerts, 2000; Lloyd, 2005). However, some studies still indicate that deterministic approaches accomplished better results and the outcomes depended on the raingage density (Dirks et al., 1998).
- Rainfall network design have been proposed in serveral studies (Ridolfi et al., 2011; Yeh et al., 2011; Cheng et al., 2012; Chen et al., 2008). A method composed of kriging and entropy is found to have certain limitations even it can be successfully applied to redesign a rainfall network of the catchment of the Shimen reservoir in Taiwan (Yeh et al., 2011; Chen et al., 2008). Ridolfi et al. (2011) present an evaluation of the rainfall network of the metropolitan area of Rome using entropy and define an empirical method to assess the maximum non-redundant information. Cheng et al. (2012) show that at least seven raingages

produce areal rainfall accurately over a 2,726 km<sup>2</sup> catchment in Taiwan. However, these methods are validated for areal rainfall or rainfall volume estimation. Our study focuses on the peak discharge simuated by a fully distributed hydrological model, based on long series of rainfall.

Indeed, the validations of rainfall interpolation methods are frequently accomplished using cross validation approaches through

- evaluation of some rainfall statistics such Root Mean Square Error (RMSE). However rainfall statistics from raingage cross validation alone can not be compared on a like-for-like basis: a better test of a rainfall interpolator for hydrological modelling is to use their rainfall estimates as model input and to assess the modelled flows against observations (Cole and Moore, 2008). Therefore, a distributed hydrological model (SIMGRO) has been used to investigate the effect of spatial variability of daily rainfall on soil moisture, groundwater level and discharge (Schuurmans and Bierkens, 2007). Ruelland et al. (2008) examined
- the sensitivity of a lumped and semi-distributed hydrological model (Hydrostrahler) to several interpolation methods of rainfall such as Thiessen polygon, inverse distance weighting, thin smooth plate splines and Ordinary Kriging (ORK). Most recently, the semi-distributed hydrological model SWAT was used to compare its performance under standard precipitation input (Thiessen polygon) and modified areal precipitation input obtained through spatial interpolation Inverse Distance and Elevation Weighting (IDEW) method (Masih et al., 2011), and regression-based interpolation (Wagner et al., 2012). Tobin et
- al. (2011) present comparative study on the IDW, ORK and KED. Then they focused on the hourly precipitation fields in complex Alpine terrain. Incorporation of KED precipitation input into a hydrological model (GSM-SOCONT) is found to provide vastly improved outputs with respect to measured discharge volumes and flood peaks for some flood events. However, the long term modeling could be a reliable validation of such method. Particularly, it should be required to draw special attention on the estimation of discharge extreme modelled using interpolated rainfall as input.
- The investigation in the present paper prolongs the work described in Ly et al. (2011). In the current paper, the objective is to model extreme discharge response to the geostatistical interpolation methods of daily rainfall, using the physically-based model EPIC-GRID. The geostatistical algorithms are developed to compute daily gridded rainfall of 30 years (1976-2005). The multivariate geostatistics are also used by incorporating elevation to recover the rainfall estimate. These geostatistical approaches are compared to inverse distance weighting and Thiessen polygon methods which are often the default methods
- implemented in various hydrological models. Furthermore, this work aims at investigating the impact of different raingage densities and positions used for interpolation, on the peak flow. This provides insight in terms of the capability and limitation of the geostatistical methods. In perticular, we try to address the question: what is the best raingage position for rainfall inputs for the extreme discharge modelling at the catchment scale in case of very sparse raingage data.

#### 2 Material and Methods

This section briefly presents the study area and main data used in this paper. The very brief description of the EPIC-GRID model is then provided. The preparation of rainfall scenarios is presented, followed by the brief description of the method for the analysis of discharge extremes. Interested readers should refer to Ly et al. (2011) for a detailed presentation of different

kriging algorithms. A detailed presentation of geostatistical theories can be found in Cressie (1991); Chilès and Delfiner (1999); Goovaerts (1997); Webster and Oliver (2007).

### 2.1. Study area and dataset

- This research is carried out in the Ourthe and Ambleve catchments situated in the Ardennes hilly landscape in the Walloon 5 region, the south-eastern part of Belgium (Figure 1A). The total area is 2908 km<sup>2</sup>, lies between 67 and 693 m in elevation (Figure 1B). The Ourthe River, a 271 km long, is a right tributary to the Meuse River. The Ourthe River is formed at the confluence of the Western Ourthe and the Eastern Ourthe, west of Houffalize. The source of the Western Ourthe is near Libramont-Chevigny and the one of the Eastern Ourthe is near Gouvy, both in the Luxembourg province, close to the border with the Great-Duchy of Luxembourg. After the confluence, the Ourthe flows roughly in north-west direction, joining the
- 10 Meuse River at Liege. The Ambleve River has its source at Honsfeld in the Bullange commune, at an elevation about 600 m, flowing into the Ourthe River near Comblain-au-Pont after a 101 km course (Figure 1B). The Ourthe flows over the banks in winter (CRO., 2005). Depending on the amounts and the localization of the precipitation, numerous settlements along the valley are excessively vulnerable to more or less large-scale flooding and damage of propterty (CRO., 2005).
- The observed discharge at all gauging stations in these catchments (Figure 1B) are given free of charge by Walloon Public Service of Hydrological Studies (SETHY, Belgium). For each gauging station, we delineated all sub-catchments boundaries using DEM provided by ERRUISSOL project. Their characteristics are presented in Table 1. The whole catchment at Sauheid gauging station has a mean altitude of 394 m, ranging from 67 m to 693 m. The Ourthe sub-catchment at Tabreux station located lower part of the area, which is assumed as low elevation catchment. The Ambleve sub-catchment located at upper part of the area, which is assumed as high elevation catchment. The Ourthe catchment at Hotton station located at the upstream
- part of the area, which is assumed as high elevation catchment.

#### 2.2. Hydrological model EPIC-GRID and validation criteria

EPIC-GRID model is extended from the original EPIC model (Williams et al., 1984) to the catchment scale and regional scale. This model is a physically-based, fully distributed, hydrological model that works on a daily time step (Figure 2). EPIC-GRID was applied to measure the effect of management on water, soil, crop growth, erosion, pesticide and nutrient fluxes in the root

- zone and the vadose zone at regional scale (Sohier et al., 2009; Sohier and Degre, 2010; Sohier, 2011). The model is able to continuously simulate the dissolved and particulate elements in large complex catchments with varying weather, soils and management conditions over long time periods. EPIC-GRID can simulate different catchment size by discretising the area into regular-squared grids. The grid size is selected according to diverse criteria such as the data availabilities and the size of simulation area that determine calculation time. Therefore, as regards to the application of the model to Walloon region
- (Belgium), the grid size retained is 1 km<sup>2</sup>. Each grid is subdivided into hydrological response units (HRUs), taking into account land use, slope, soil and meteorological data. EPIC-GRID linked up to Geographic Information System (GIS) can integrate various spatial environmental data, including soil, land cover, climate and topographical features. In each HRU, the simulations

are carried on at a daily time step from 1961 to 2005. In this work, the results are taken into account from 1976 to 2005. Eventually, the outputs of a grid cell are a weighted sum of results of the HRUs, considering their percentage of surface in the grid as weighting factor. The flows can then be aggregated on the basis of each reporting unit or they can be used as a raster file into a GIS.

- The EPIC-GRID model has been validated for all catchments of the Walloon region (Sohier et al., 2009; Sohier and Degre, 5 2010). For full details on the description, modification and validation of the EPIC-GRID model, we refer to Sohier (2011). In this study, the validation of the EPIC-GRID model is assessed by two quantitative indexes Nash-Sutcliffe efficiency (NSE) and per cent bias (PBIAS). NSE is a normalized statistic that determines the relative magnitude of the residual variance compared to the measured data variance. NSE indicates how well the plot of observed versus simulated data fits the 1:1 line
- 10 (Moriasi et al., 2007). NSE is calculated as followed (Moriasi et al., 2007):

$$NSE = I - \frac{\sum_{i=1}^{n} (Qobs_i - Qsim_i)^2}{\sum_{i=1}^{n} (Qobs_i - Qmean)^2}$$
(1)

where Qobsi is the ith observed discharge, Qsimi is the ith simulated discharge, Qmean is the mean of observed discharge, and n is the total number of observations.

Major disadvantage of NSE is that, even if a model systematically over- or underestimates observations (OBS) all the time, the NSE still provides good values close to 1.0 (Moriasi et al., 2007). As a result, PBIAS can overcome this drawback. PBIAS 15 measures the average tendency of the simulated discharge to be larger or smaller than the observed discharge. The optimal value of PBIAS is 0.0, with low-magnitude values indicating accurate model simulation. Positive values indicate model underestimation bias, and negative values indicate model overestimation bias (Moriasi et al., 2007). PBIAS is computed as followed:

20 
$$PBIAS = \frac{\sum_{i=1}^{n} (Qobs_i - Qsim_i) \times 100}{\sum_{i=1}^{n} Qobs_i}$$
(2)

where PBIAS is the deviation of data being evaluated, expressed as a percentage. The simple average of residues can give the same indication as for over- or under-estimate of the simulated flows, which is directly interpretable. However, the PBIAS is recommended by Moriasi et al. (2007) as it is commonly used to evaluate the model simulation (Fernandez et al., 2005; Gupta et al., 1999; Singh et al., 2005). PBIAS can quantify water balance errors, its use can easily be extended to load errors and it has the ability to clearly indicate poor model performance.

Natural Hazards and Earth System Sciences Discussions

Through the guideline established and recommended by Moriasi et al. (2007), the model result is very good if  $0.75 

5

analysis. The Grubbs and Beck (1972) test (G-B) was established to detect outliers. Then, the outliers are removed before the analyses. The test procedure can be found in Rao and Hamed (2000).

The annual maxima are ordered and plotted using Cunnane plotting position (Chow et al., 1988). Plotting position refers to the probability value assigned to each piece of data to be plotted. Numerous methods have been proposed for the determination of the plotting positions, most of which are empirical. If n is the total number of values to be plotted and m is the rank of a value in a list ordered by descending magnitude, the exceedance probability of the mth largest value, xm, is for large n, can be

$$P(X \ge x_m) = \frac{m-b}{n+1-2b}$$

And the return period:  $T = \frac{1}{P(X \ge x_m)}$ 

generally represented by:

- 10 where b is a parameter: b = 0.5 for Hazen's formula, b = 0.3 for Chegodayev's, b = 0 for Weibull's, b = 3/8 for Bloom's, b = 1/3 for Tukey's, and b = 0.44 for Gringorten's (Chow et al., 1988). The Weibull plotting formula is biased and plots the largest values of a sample at too small a return period (Cunnane, 1978). The research on the plotting positions has had long history and the work is still continuing (Rao and Hamed, 2000). In the HYFRAN© package, the default value of b is 0.4 corresponds to the Cunnane plotting position which is chosen for the analysis in this study.
- Once the data series are identified and ranked, and the plotting positions calculated, a graph of magnitude (x) vs. probability [(P(X>x), P(X

5

best adjustments correspond to the higher values of a posteriori probability. Additional information was also given: the lowest values of Bayesian information criterion (BIC) and Akaike information Criterion (AIC). The five best-classified functions are retained and the hypothesis test (Chi-Squared) is applied in order to check the adequacy of these functions to the sample of observed values. The choice of the best function is made in a visual way by analysis of the graph representing five best classified fitting (Dautrebande et al., 2006). The examination of the graph involves of selecting the distribution function that

characterizes the most reliable with the data series. The selected distribution should be fitted well at higher return period. The 1- or 2-parameter distribution is always desired except the adjustment quality of a distribution of 3 parameters is sharply superior to that of the distributions of 2 parameters.

#### 3. Results and Discussion

10 At the main outlets, the analyses of extreme discharge is carried out for annual maximum series of observed discharges and simulated discharges modelled using EPIC-GRID model with different rainfall scenarios interpolated using different raingage densities and positions.

### 3.1. Validation of the EPIC-GRID model

As the objective of this paper is to model the extreme discharge response to the geostatistical interpolation methods of daily rainfall, we show only the result of the model validation as shown in Table 2. Further discussion is another subject which will be reported elsewhere.

#### 3.2. Impact of interpolation methods on extreme discharge

In general for the case of using all available raingages, IDW is not always the best method to capture rainfall in order to simulate good peak discharge. The methods that take into account elevation as external data and co-variables like KED and

20 OCK respectively, can improve the rainfall to produce very good simulated peak discharge at high elevation sub-catchment (Figure 4). Nonetheless, all scenarios are in good agreement as they produce simulated peak discharge and distribution functions that are in the area of 95%-confidence interval of observed distribution

In Figure 4, four graphs show the extreme peak discharge of the observed data and the different simulated scenarios as a function of their return periods for four outlets in the Ourthe and Ambleve catchments. The discharges at 100 year return period

- 25 observed at each gauging station and simulated by different interpolated rainfall using 70 raingages are shown in Table 3. The peak discharges and their distributions of all scenarios are generally overestimated for lower return periods. At higher return periods, for the whole catchment at Sauheid outlet, the peak discharge modelled using ORK rainfall scenario follows very closely the distribution function of observed data. The OCK scenario is slightly overestimated. The distribution functions of other four scenarios are superposed each other and stay lower than the distribution function of observed discharge. Always at
- 30 the higher return periods, for the low sub-catchment at Tabreux outlet, the simulated peak discharges are lower than observed

from observation.

5

peak discharge and its distribution function. However, the distribution of simulated peak discharge using OCK is closest to observed distribution while the one of THI is the lowest and far from observed distribution function.

For the smaller and highest elevation sub-catchment at Martinrive outlet, the peak discharge simulated using KED and OCK follow the observed peak discharge very closely. The THI scenario has slightly higher peak discharge and its distribution function. The UNK and ORK scenarios have quite the same peak discharge and distributions that are lower than observation. The IDW scenario stays lowest and far from observation. At small and high sub-catchment at Hotton outlet (lower than Martinrive), all simulated peak discharge and their distributions are lower than observed peak discharge and its distribution. But KED and OCK still provide closest simulated peak discharge and distribution and the IDW scenario is still lowest and far

#### 10 **3.3. Impact of raingage densities on extreme discharge**

All scenarios still are in good agreement as their simulated peak discharge and distribution functions are in the area of 95%confidence interval of observed distribution. The differences in simulated peak discharges when using 8 raingages are not clearly distinguished from the differences when using all available raingages. The discharges at 100 year return period observed at each gauging station and simulated by different interpolated rainfall using 8 raingages are shown in Table 4. At low return

- periods, the simulation is still higher than observation for both peak discharges and fitted distribution functions (Figure 5). However, the interpolation method ranking is changed at higher return periods. The UNK scenario has lowest peak simulated discharge and its distribution than observation's one while other scenarios follow closely the observation's distribution for the whole catchment at Sauheid outlet. Always at high return period for smaller sub-catchment at low elevation part (Tabreux outlet), three methods (THI, KED and UNK) have low peak discharge and distribution functions while IDW, ORK and OCK
- follows very closely the observation's distribution. Also for smaller sub-catchment but at very high elevation part (Martinrive outlet), the differences in simulated peak discharges and distributions appear clearly large (Figure 5). The IDW and ORK follow very closely the observed peak discharge and distribution function. The UNK and OCK have slightly higher and lower respectively than observed distribution. The THI and KED have very low simulated peak discharge and distribution functions. For the smallest sub-catchment but relatively lower elevation part (Hotton outlet), the OCK has the simulated peak discharge
- and distribution closest to the observed one. The UNK has the lowest simulated peak discharge and distribution. Other fours methods have simulated peak discharge and distribution slightly lower than observed one. For this raingage density case, the IDW, ORK and OCK follow generally closely the observations. The THI and KED have the observed peak discharge and distribution very close to the observed one for the whole catchment but they are not very appropriate for sub-catchments.

#### 30 **3.4. Impact of raingage positions on extreme discharge**

For very scarce raingage cases, the raingage position is the main issue defining the rainfall prediction and hence the simulated peak flows at the outlets in the catchment area. In the present paper, five cases of using four raingages are degenerated for

interpolation rainfall as input for hydrological modelling. In spite of the same number of raingage, the modelling results are considerably dissimilar.

The first position of 4 raingages, the 4a raingages are located around the catchments as shown in Figure 3. In Figure 6, four graphs display the extreme peak discharge and distribution functions of the observed data and the different scenarios according

- to their return periods for four outlets in the catchment area. The discharges at 100 year return period observed at each gauging station and simulated by different interpolated rainfall using 4a raingages are shown in Table 5. For all outlets, the simulated peak discharge and distribution functions are always overestimated for low return periods. For the whole catchment, the model using rainfall scenarios from this raingage position case has comparable simulated peak discharge and distributions at higher return period, compared to the higher raingage density cases. Nevertheless, the differences in simulated peak discharge and
- distributions at higher return period look obviously large for the sub-catchments. The rainfall scenarios of all methods provide overestimated peak discharges for the low sub-catchments (Figure 6). The UNK and KED have the simulated peak discharge and distribution functions very far from the observed ones and quite

outside the 95% confidence interval of the observed distribution function. In contrast for the high sub-catchment, the differences in simulated peak discharge and distributions are relatively large. This can also be explained by the distance

between the raingage polygon and the sub-catchment positions. The difference in extreme discharge becomes larger when the distance increases (Figure 6).

It is also remarkable that OCK and UNK produce the simulated peak discharge and the distribution functions very close to the observed ones (Figure 6). The KED still has the simulated peak discharge and the distribution function far from the observed distribution function.

- Other raingage positions have also been implemented for the same number of raingage to determine which raingage position can improve the spatial rainfall input for modelling of extreme discharges. The 4b raingages are located as shown in Figure 3. Three raingages are outside the catchment area and mostly at downstream part of the Ourthe and one raingage is at the source of the eastern Ourthe. The discharges at 100 year return period observed at each gauging station and simulated by different interpolated rainfall using 4b raingages are shown in Table 6. In this case, the difference in peak discharges simulated using
- rainfalls of the different interpolation methods is relatively large (Figure 7). The ORK has the simulated peak discharge and distribution closely to the observed ones for low sub-catchment (Tabreux outlet). But it provides worsen simulated peak discharge and distribution functions for smaller and higher sub-catchments, especially in the highest sub-catchment, the simulated peak discharge and distribution is outside the area of 95% confidence interval of the observed distribution. It is remarkable that the KED distribution follow quite well the observed one for higher sub-catchments. Other interpolation
- methods have somewhat their distributions inside the 95% confidence interval of the observed distribution. It is also shown that the difference in extreme discharge generated by different rainfall becomes larger according to the increased distance between the centroid of the raingage polygon and the catchments (Figure 7).

It seems that the simulated peak discharge by UNK is discontinuous for the Tabreux outlet but it is not. This is due to the fact that peak discharge simulated by UNK is strangely high for several years and cause large difference with the previous data of the sorted years. This is really the disadvantage of the UNK in the interpolation process.

- The 4c raingages are located as shown in Figure 3. Two raingages are close to each other and located at the source of the 5 western Ourthe. One raingage is at low elevation, near the Meuse River and outside the catchment. The last raingage is also outside the catchment and located at Ambleve catchment side. In this case, the differences in peak discharges simulated using rainfalls of the different interpolation methods are also large, especially for the sub-catchments. It is also shown that the difference varies according to distance between the centres of the raingage polygon and the catchments. The two methods UNK and KED have their distribution functions that tend to be outside the 95% confidence interval of the observed distribution
- (Figure 8). Other methods have their simulated distribution functions inside the 95% confidence interval of the observed distribution even if some simulated distribution functions do not follow the observed distribution very close for the highest sub-catchment.

Other raingage positions have been conducted for the same number of raingage. The 4d raingages are located as shown in Figure 3. The raingages are only at the higher part of the catchment area. In this case, the differences in peak discharges

- modelled using rainfalls of the different interpolation methods become very large. The UNK and KED are significantly different from the other four methods. Their simulated peak discharges and distribution functions are outside the 95% confidence interval of the observed distribution (Figure 9). Exception for the highest sub-catchment, UNK distribution follows closely the observed distribution even if the distance between the centroids of the raingage polygon and the sub-catchment is farthest. This is because the raingage polygon still covers the big part of the sub-catchment.
- In overall for higher return periods, it is noticeable that four methods (THI, IDW, ORK and OCK) provide quite good model performance for this raingage case. The UNK and KED cannot provide good simulated peak discharge (Figure 9). This is probably because of the raingage coordinates of this configuration that are used as external drift in the UNK and KED. Using elevation alone as OCK can produce quite good simulated peak flow.

The last case of very sparse raingages is used for interpolation of rainfall, the difference in simulated peak discharges also become larger. The 4e raingages are located only at low part of the whole catchment (Figure 3). This case provides very large difference in peak flow modelled using rainfall input from the interpolation methods (Figure 10). The KED distributions are all outside the 95% confidence interval of observed distributions. The UNK distributions are outside the 95% confidence interval of observed distributions for the whole catchment and the highest sub-catchment. At the low sub-catchment, the UNK distributions follow closely the observed distributions. This is also influenced by the distance between the centroids of the

30 raingage polygon and the catchments. The centroid of low sub-catchment is near the centroid of the raingage polygon. The distributions of other four methods (THI, IDW, ORK and OCK) are inside the 95% confidence interval of observed distributions for the whole catchment and the two low sub-catchments. But all are outside the 95% confidence interval of observed distributions for the highest sub-catchment (Figure 10).