# Peer review of "Modelling extreme discharge response to several geostatistically interpolated rainfall using very sparse raingage data"

_Natural Hazards and Earth System Sciences, 2016_

## Referee Comment (RC1) · Anonymous Referee #1 · 1 Mar 2016

General Comments

The paper deals with a very interesting topic based on the relationship between simulated river flows and rainfall interpolation procedures. Then, some recommendations to interpolate rainfall would be elucidated. Specifically when disperse rain gauges are available. But this is also a very difficult problem because of mixing so many uncertainties. At least those related with the suitability of rain gauge network to estimate rainfall maps, the goodness of multivariate procedures to interpolate rainfall and the parameterization of a distributed hydrological model. Furthermore, they focused on extreme events simulated with a daily continuous hydrological model which seems to be another difficult task. After reading the paper I would recommend a major revision

of it because of: 1. There are so many techniques described that obscured what it may be their principal analysis: "... effect of different raingage densities and particularly the effect of the raingage positions for very sparse raingage data used for rainfall interpolation, on extreme flow". Note the ambitious of this study that encompasses rainfall interpolation, the analysis of rain gauge network, the use of a continuous hydrological model and the assessment of maximum discharges. 2. Conclusions aren't derived from a detailed analysis of results. My opinion is that the description of results is only descriptive of some specific simulations (scenarios) but causes aren't analyzed and authors don't offer clues to take into account in other basins 3. The complexity of the topic makes almost impossible to differentiate what is due to interpolation methodology and what is derived from hydrological modelling. Why the authors didn't use a cross validation technique to elucidate which are general recommendations for a multivariate interpolation procedure of rainfall in order to clarify their conclusions? 4. Main findings related to kriging are obscure and not directly useful. Authors remarked the importance of rain gauge position but they didn't offer insights of different rainfall samples and their representativeness of rainfall in each basin 5. An index is proposed to " ... illustrate the quality of the raingage distribution with respect to the calculation of extreme discharge" but I found it that it wasn't properly described. Authors neither used a mathematical formula nor analyzed its sources of variability, i.e. its domain of values and their significance

Addressing scientific questions

Some scientific questions may be proposed to authors: 1. According to the use of multivariate methodologies, are there any sources of variability in rainfall and elevation relationships (temporal or spatially variation) that may affect parameterization of kriging and subsequently those methods compared? 2. What are the advantages of addressing the suitability of interpolation procedures by means of discharge extremes instead of using a cross validation procedure 3. Wouldn't be profitable to identify sources of uncertainty in your work? 4. There is an index proposed to describe how suitable is a rain

gauge network for hydrological modelling. But it doesn't work with elevation or aspect values that it is recognized to influence rainfall. Wouldn't it be a major disadvantage to use the index proposed?

Technical corrections

• Whatever methodology is used or selected to interpolate rainfall, it can be objectively parameterized. No matter if it is deterministic or multivariate. Cross validation is a well know methodology that allows the estimation of errors and the optimization of interpolation procedures. • I would recommend to use only one term to refer ground rain gauges: rain gauge, raingages or weather station • Maximum elevation in analyzed basins is 693 m which can't represent high elevations. What is the variability of recorded rainfall? • Daily records of rainfall, are free of errors and gaps?. • • The removing of outlier discharge is based on annual maximum daily analysis. As stated in the paper, outliers are also due to natural causes. So, their removal is an open question that can be discussed. I also wonder how the removal of such kind of data can affect to a continuous hydrological modeling • How rainfall scenarios are designed and why are based on the selection of 70, 8 and 4 rain gauges. The same question for the positions. How are they selected? • Can't see rain gauge positions in Figure 3 • Figure 4 is not clear because of line thickness • Wouldn't be useful to express mathematically the index used to describe the performance of the rain gauges in a basin?. And then, what are usual values, over what range would you say that rain gauge network is suitable for hydrological modelling, what are the criteria to select rain gauges to work with. How can we refer this index

Typing errors

General Recommendation: review of English Line 30, page 2: "serveral"Âů Line 2, page 3: "simuated"Âů

---

## Referee Comment (RC2) · Anonymous Referee #2 · 14 Mar 2016

This manuscript presents a study in which the value of different rainfall interpolations was assessed by hydrological modeling. While this is an interesting topic and a valuable approach, I am afraid I have major concerns with the manuscript in its current form, mainly because so much remains unclear in the presentation. I find this a pity, because I really like the approach and it is rather unfortunate that the authors were not able to present their work in a better way.

Mainly, I had problems to fully understand what actually had been done. The hydrological model is a central part of the study, but very little information on this is given. Important questions to me include: How has the model been parameterized? Was there any calibration involved? If yes, how? What was the temporal resolution of the

runoff model in this study? How well did the model perform for extreme events? How well were internal variables reproduced? With the central role the hydrological model plays in this study, the results cannot be fully understood and interpreted, it this information is missing!

Even after reading the manuscript several times, it remained unclear what exactly had been done in this study. This makes it basically impossible to assess the manuscript. As a reviewer you do not want to have to guess the methods. This also means that I was not able to really review the study.

Partly rather basic things are described in great detail (e.g. p 7 , plotting position), but then at other places important detail information is missing. The method section needs a major revision to allow understanding what actually has been done. I also found section 3 hard to follow, the mix between results and discussion is just confusing!

While the authors are right about the limitation of the model efficiency it is not too helpful to just compute the volume error (here called PBIAS) does not solve the issue. Rather one should use some combined measure (e.g., Lindström 1997)

The equations are poorly written (NSE is no suitable variable name!)

The text switches (randomly) between present and past tense (as example see section 2.4). Also in other aspects, the manuscript would largely benefit from improving the quality of the language.

Lindström, G., A simple automatic calibration routine for the HBV model. Nordic Hydrology, 28: 153-168, 1997.

---

## Author Comment (AC1) · 19 May 2016

Dear Referee #1,
We are very grateful for your constructive review of our paper. We will try to take advantage of your advice for improving the manuscript.

General Comments
The paper deals with a very interesting topic based on the relationship between simulated river flows and rainfall interpolation procedures. Then, some recommendations to interpolate rainfall would be elucidated. Specifically when disperse rain gauges are available. But this is also a very difficult problem because of mixing so many uncertainties. At least those related with the suitability of rain gauge network to estimate rainfall maps, the goodness of multivariate procedures to interpolate rainfall and the parameterization of a distributed hydrological model. Furthermore, they focused on extreme events simulated with a daily continuous hydrological model which seems to be another difficult task. After reading the paper I would recommend a major revision of it because of: 1. There are so many techniques described that obscured what it may be their principal analysis: ". . . effect of different raingage densities and particularly the effect of the raingage positions for very sparse raingage data used for rainfall interpolation, on extreme flow". Note the ambitious of this study that encompasses rainfall interpolation, the analysis of rain gauge network, the use of a continuous hydrological model and the assessment of maximum discharges. 2. Conclusions aren't derived from a detailed analysis of results. My opinion is that the description of results is only descriptive of some specific simulations (scenarios) but causes aren't analyzed and authors don't offer clues to take into account in other basins 3. The complexity of the topic makes almost impossible to differentiate what is due to interpolation methodology and what is derived from hydrological modelling. Why the authors didn't use a cross validation technique to elucidate which are general recommendations for a multivariate interpolation procedure of rainfall in order to clarify their conclusions? 4. Main findings related to kriging are obscure and not directly useful. Authors remarked the importance of rain gauge position but they didn't offer insights of different rainfall samples and their representativeness of rainfall in each basin 5. An index is proposed to " . . . illustrate the quality of the raingage distribution with respect to the calculation of extreme discharge" but I found it that it wasn't properly described. Authors neither used a mathematical formula nor analyzed its sources of variability, i.e. its domain of values and their significance

We were trying to shorten the paper, that's why all methods are very briefly described but the references are precisely cited, if the readers need to know details about the methods. However, we will make clearer about the techniques details in the revised manuscript. We respond below to your scientific questions item by item:

**Addressing scientific questions**
Some scientific questions may be proposed to authors:
1. According to the use of multivariate methodologies, are there any sources of variability in rainfall and elevation relationships (temporal or spatially variation) that may affect parameterization of kriging and subsequently those methods compared?
We used the elevation information extracted from Digital Elevation Model (DEM) in order to improve the estimation by using the Kriging with an External Drift (KED) and Ordinary Cokriging (OCK). The KED used the elevations as the secondary variable to derive the local mean of rainfall (primary variable) while OCK took advantage of the correlation between the two variables (elevation and rainfall). Over the 30 yr (1976–2005), there were a total of 10063 rain days, on which the Pearson's coefficient was computed from the correlation between rainfall amount and elevation. Of these 10063 rain days, 2087 rain days (20.74 %) had a Pearson's correlation coefficient higher than 0.5 and 181 rain days (1.8 %) had a Pearson's correlation coefficient lower than −0.5. This analysis is already done and published in our previous paper (Ly et al 2011). We think that this analysis does not affect the parameterization of kriging.

2. What are the advantages of addressing the suitability of interpolation procedures by means of discharge extremes instead of using a cross validation procedure?
As stated in the manuscript, the evaluation of interpolation methods are frequently accomplished using cross validation approaches through evaluation of some statistics. However cross validation alone can not be compared on a like-for-like basis. Here, we proposed a better test of a rainfall interpolator for hydrological modelling that is to use their rainfall estimates as model input and to assess the modelled flows against observations. Moreover, we think that it is an innovative idea to compare by taking more attention on extreme discharge analysis.

3. Wouldn't be profitable to identify sources of uncertainty in your work?
We will try to address the sources of uncertainty in our revised manuscript.

4. There is an index proposed to describe how suitable is a rain gauge network for hydrological modelling. But it doesn't work with elevation or aspect values that it is recognized to influence rainfall. Wouldn't it be a major disadvantage to use the index proposed?

It is recognized that elevation and aspect have influence on rainfall. But the proposed index (our result confirm that) is one of the key factors to define the model's performance. We will discuss this point in the revised paper.

Technical corrections

âˇA ´c Whatever methodology is used or selected to interpolate rainfall, it can be objectively parameterized. No matter if it is deterministic or multivariate. Cross validation is a well know methodology that allows the estimation of errors and the optimization of interpolation procedures. âˇA ´c I would recommend to use only one term to refer ground rain gauges: rain gauge, raingages or weather station âˇA ´c Maximum elevation in analyzed basins is 693 m which can't represent high elevations. What is the variability of recorded rainfall? âˇA ´c Daily records of rainfall, are free of errors and gaps?. âˇA ´c âˇA ´c The removing of outlier discharge is based on annual maximum daily analysis. As stated in the paper, outliers are also due to natural causes. So, their removal is an open question that can be discussed. I also wonder how the removal of such kind of data can affect to a continuous hydrological modeling âˇA ´c How rainfall scenarios are designed and why are based on the selection of 70, 8 and 4 rain gauges. The same question for the positions. How are they selected? âˇA´c Can't see rain gauge positions in Figure 3 âˇA ´c Figure 4 is not clear because of line thickness âˇA ´c Wouldn't be useful to express mathematically the index used to describe the performance of the rain gauges in a basin?. And then, what are usual values, over what range would you say that rain gauge network is suitable for hydrological modelling, what are the criteria to select rain gauges to work with. How can we refer this index

Typing errors
General Recommendation: review of English Line 30, page 2: "serveral"Â °u Line 2,
page 3: "simuated"°u

We thank you very much for technical corrections that help us to improve our paper. We will address the parameters that we use in the interpolation methods. Please refer to answer #2 for the issue of cross validation.

We will revise the terms that refers to ground raingages.

The rainfall varied according to the elevation as explained in the answer # 1. A basic rainfall error detection is already included in our algorithms and the rainfall data itself have usually gaps, missing data that we did not consider in the calculation.

We made an outlier test based on Rao and Hamed, 2000 for the simulated discharge for the statistics of extreme, so it does not affect to the continuous hydrological modeling.

The rainfall scenarios are design based on available raingages (70) in and surrounding the catchment. Then we reduce the number of raingages and the raingages are randomly chosen.

Figure 3 and 4 is pretty clear according to the file, maybe it is because of printing problem, and anyway we will improve it.

We will put more information on index of position to make the results more understandable. We will recommend the range of the index. According to our results, we can say that the index below 1.5 is suitable for hydrological modelling.

We will improve English of the manuscript.

**References**:
Ly, S., Charles, C., and Degre, A.: Geostatistical interpolation of daily rainfall at catchment 556 scale: the use of several variogram models in the Ourthe and Ambleve catchments, Belgium, 557 Hydrol Earth Syst Sc, 15, 2259-2274, 10.5194/hess-15-2259-2011, 2011.

Rao, A. R., and Hamed, K. H.: Flood frequency analysis, 1st ed., CRC Press, Florida, 376 pp., 570 2000.

---

## Author Comment (AC2) · 19 May 2016

Dear Referee #2,
We are very grateful for your constructive review of our paper. We will try to take advantage of your advice for improving the manuscript. We respond below to your comments item by item:

This manuscript presents a study in which the value of different rainfall interpolations was assessed by hydrological modeling. While this is an interesting topic and a valuable approach, I am afraid I have major concerns with the manuscript in its current form, mainly because so much remains unclear in the presentation. I find this a pity, because I really like the approach and it is rather unfortunate that the authors were not able to present their work in a better way.

We were trying to shorten the paper, that's why all methods are very briefly described but the references are precisely cited, if the readers need to know details about the methods. However, we will make clearer about the details techniques in the revised manuscript.

Mainly, I had problems to fully understand what actually had been done. The hydrological model is a central part of the study, but very little information on this is given. Important questions to me include: How has the model been parameterized? Was there any calibration involved? If yes, how? What was the temporal resolution of the runoff model in this study? How well did the model perform for extreme events? How well were internal variables reproduced? With the central role the hydrological model plays in this study, the results cannot be fully understood and interpreted, it this information is missing!

The important and concise information of the hydrological model have been presented in the manuscript while the application of this model in the region has also been published in several papers (Sohier et al., 2009; Sohier and Degre, 2010; Sohier, 2011). EPIC-GRID model is extended from the original EPIC model by Williams et al., 1984 to the catchment scale and regional scale. The EPIC-GRID model has been parametrized, calibrated and validated for all catchments of the Walloon region (Sohier et al., 2009; Sohier and Degre, 2010). For full details on the description, calibration and validation of the EPIC-GRID model, we refer to Sohier (2011).

As mentioned in the Line 157 of the manuscript, the model is a physically-based, fully distributed, hydrological model that works on a **daily time step** for runoff simulation.

The revised manuscript will present performance indicators or the model in order to help the reader to understand and interpret the results.

Even after reading the manuscript several times, it remained unclear what exactly had been done in this study. This makes it basically impossible to assess the manuscript. As a reviewer you do not want to have to guess the methods. This also means that I was not able to really review the study.
We will present clearer about the method in the revised manuscript.

Partly rather basic things are described in great detail (e.g. p 7 , plotting position), but then at other places important detail information is missing. The method section needs a major revision to allow understanding what actually has been done. I also found section 3 hard to follow, the mix between results and discussion is just confusing!
We will revise the manuscript to make it understandable and we will separate the result and discussion sections.

While the authors are right about the limitation of the model efficiency it is not too helpful to just compute the volume error (here called PBIAS) does not solve the issue. Rather one should use some combined measure (e.g., Lindström 1997).
We use the model evaluation guidelines for systematic quantification of accuracy in watershed simulations by Moriasi et al., 2007 which is most recent publication.

The equations are poorly written (NSE is no suitable variable name!)
We will check the equations but in this case we think it is correct. We can rename NSE.

The text switches (randomly) between present and past tense (as example see section 2.4). Also in other aspects, the manuscript would largely benefit from improving the quality of the language.

We will improve the quality of the language.

**References:**

Moriasi, D. N., Arnold, J. G., Van Liew, M. W., Bingner, R. L., Harmel, R. D., and Veith, T. 567 L.: Model evaluation guidelines for systematic quantification of accuracy in watershed 568 simulations, T Asabe, 50, 885-900, 2007.

Sohier, C., Degre, A., and Dautrebande, S.: From root zone modelling to regional forecasting 591 of nitrate concentration in recharge flows - The case of the Walloon Region (Belgium), J 592 Hydrol, 369, 350-359, DOI 10.1016/j.jhydrol.2009.02.041, 2009. 593

Sohier, C., and Degre, A.: Modelling the effects of the current policy measures in agriculture: 594 An unique model from field to regional scale in Walloon region of Belgium, Environ Sci 595 Policy, 13, 754-765, DOI 10.1016/j.envsci.2010.08.008, 2010.

Sohier, C.: Développement d'un modèle hydrologique sol et zone vadose afin d'évaluer 597 l'impact des pollutions diffuses et des mesures d'atténuation sur la qualité des eaux en région 598 wallonne, Ph.D thesis, Gembloux Agro-Bio Tech, University of Liege, Gembloux, 338 pp., 599 2011.